# Trends in Harnessing Plant Endophytic Microbiome for Heavy Metal Mitigation in Plants: A Perspective

**DOI:** 10.3390/plants12071515

**Published:** 2023-03-31

**Authors:** Pragya Tiwari, Hanhong Bae

**Affiliations:** Department of Biotechnology, Yeungnam University, Gyeongbuk, Gyeongsan 38541, Republic of Korea

**Keywords:** defense biome, endophytes, ‘heavy metal resistome’, metal bioavailability, stress mitigation, environment sustainability

## Abstract

Plant microbiomes represent dynamic entities, influenced by the environmental stimuli and stresses in the surrounding conditions. Studies have suggested the benefits of commensal microbes in improving the overall fitness of plants, besides beneficial effects on plant adaptability and survival in challenging environmental conditions. The concept of ‘Defense biome’ has been proposed to include the plant-associated microbes that increase in response to plant stress and which need to be further explored for their role in plant fitness. Plant-associated endophytes are the emerging candidates, playing a pivotal role in plant growth, adaptability to challenging environmental conditions, and productivity, as well as showing tolerance to biotic and abiotic stresses. In this article, efforts have been made to discuss and understand the implications of stress-induced changes in plant endophytic microbiome, providing key insights into the effects of heavy metals on plant endophytic dynamics and how these beneficial microbes provide a prospective solution in the tolerance and mitigation of heavy metal in contaminated sites.

## 1. Introduction

Biotic and abiotic stresses constitute major threats to the ecosystem and cause constraints on food security, significantly hampering crop productivity worldwide. In addition to biotic stress, abiotic stresses adversely impact plant physiology, functions and productivity, affecting plant growth and yield by up to 50 and 70% in most plant species [1]. The exposure of plants to challenging environments results in major abiotic stresses, including drought, salinity, high and low temperature, heavy metals and acidic/alkaline conditions [2]. The rising global population and climatic fluctuations are other crucial factors in providing food security to millions across the globe [3], necessitating an immediate response to tackle the growing concerns.

Plant microbiomes represent dynamic entities that are influenced by the environmental stimuli and stresses in the surrounding conditions. During their entire life cycle, plants are constantly subjected to biotic and abiotic stresses, which adversely influence plant growth, survival and adaptability to challenging conditions [4,5,6]. These changes alter plant transcriptome and metabolome via changes in leaf and root exudates, considerably impacting plant-associated microbes and their dynamics [7,8]. The plant microbiome comprises the integrally associated microbes with their plant hosts (present in endosphere) and in the rhizosphere, leaf tissue, etc., and collectively designated as the ‘plant microbiome’. The coevolution of plants with their associated microbiomes has positively impacted both associates, with microbes assisting plants in dealing with environmental stresses, delineated as the ‘cry for help’ defense employed by the plants [9,10]. In case of any type of stress, including pest-induced, abiotic, or pathogen attacks affecting plants [10,11], the beneficial microbes are recruited to promote stress tolerance [12,13,14], and these plant–microbiome dynamics are crucial for nutrient acquisition [15] and plant development [16,17].

Studies have suggested that the increased abundance of plant-associated microbes in response to stress conditions may favorably impact plant fitness and survival. In this direction, key studies have documented the potential of microbes for plant rescue in stress conditions, and the potential strains include *Pseudomonas* spp., *Chryseobacterium* spp., *Stenotrophomonas* spp., *Flavobacterium* spp., *Xanthomonas* spp., and *Chitinophaga* spp., among others, often showing enrichment in plants on exposure to microbial attacks. These microbes in association with the plant host have evolved mechanisms to aid plant defense, induce defense signaling pathways in plants [18], and restrict pathogen growth, thus ameliorating stresses on plants [19,20]. Liu and coworkers [7] proposed the concept of a ‘Defense Biome’ to better understand and manipulate plant microbiomes and counter challenging stresses. On the exposure of the plant to stress conditions, microbes associated with plants can be classified accordingly as (i) microbial taxa that remain unaltered, (ii) microbial taxa that decline in abundance, and (iii) microbial taxa that are increased in composition. These microbial consortia are collectively defined as ‘Defense Biome’ for a specific stress-influencing plant [20].

Abiotic stresses continue to adversely impact terrestrial ecosystems, further aggravated by modern agricultural practices, industrialization, and mining, deteriorating the quality of life and the environment [17,21]. The increase in heavy metal (HM) concentration in water, soil, and air is toxic for animals and plants and further contributes to environmental concerns. Unlike the presence of organic pollutants, the use of natural processes for bioremediation is not sufficient, leading to the accumulation of HM in the food chain, which has an adverse impact on the metabolism, biomass, composition, diversity, and other functions of microbial and plant communities [22].

To tackle this challenge, plant systems have evolved physiological and molecular mechanisms for stress tolerance, adaptability, and survival [23]. While advances in plant breeding and genetic engineering have witnessed translational achievements in the development of metal stress-resistant varieties, these techniques are often cumbersome and time-consuming. Plant-associated beneficial microbes show potential in addressing HM contamination, decreasing metal accumulation in plant tissues, and assisting in the reduction of the bioavailability of metals in the soil [24,25]. Endophytes are ubiquitously distributed in plant tissues and have gained global recognition in ameliorating plant stresses in recent decades. The beneficial effects of endophytes in plant rescue and aid in salinity, drought, nutrient deficiency, and pathogen infection [26,27] contexts have been extensively documented and are gaining widespread research interests in the present era. Endophytes define an attractive and sustainable bio-based approach to promote crop productivity and tolerance to environmental stresses [25]. Bacterial and fungal endophytes have demonstrated positive outcomes in addressing HM contamination and increasing plant tolerance for better adaptability and survival, as discussed in key studies [17,28]. Table 1 discusses the bioremediation potential of plant-associated endophytes in the tolerance/mitigation of HM-associated stress in plants, highlighting key examples.

The colonization of plants by endophytes comprise a natural yet complex phenomenon, and elucidation of the mechanism of plant–endophyte dynamics would substantially benefit socio-economic outcomes, including agriculture [49,50,51]. While substantial progress has been achieved in employing plant-associated endophytes in HM tolerance and mitigation [52,53] with translational success, endophytes’ efficiency in phytoremediation can be further improved via genetic manipulation and the creation of high-value endophyte strains. The synthetic biology-mediated chassis of endophytes has opened new avenues in microbial strain improvement for the desired function, whether it be enhanced secondary metabolite production [17,54], phytoremediation [55], plant growth and development, or other significant initiatives from a socio-economic perspective. The suitable plant–endophyte combination can effectively enhance plant growth and organic contaminants/pollutants degradation in the endosphere/rhizosphere [56]. In this direction, studies are directed to decipher the mechanisms of the endophyte-mediated phytoremediation of pollutants and organic contaminants in the soil. In addition, omics biology (including metagenomics, metaproteomics, metatranscriptomics, and metabolomics) has substantially contributed to the characterization of plant microbial communities, improving our understanding of plant–microbe dynamics and functional mechanisms [57]. Figure 1 highlights recent trends in harnessing plant endophytic microbiomes for HM mitigation in plants.

In this article, efforts have been made to discuss and understand the stress-induced changes in plant endophytic microbiome, providing key insights into the effects of HMs on plant endophytic dynamics and how these plant-associated beneficial microbes provide a prospective solution in the tolerance and mitigation of HMs in contaminated sites.

## 2. Heavy Metal Stress and Its Impact on Plant Endophytic Microbiome

The presence of a high concentration of HM projects adverse consequences for the health and functioning of the ecosystem, as shown by Quanzhou bay (a HM-contaminated estuary; an invasion of *Spartina alterniflora* in the estuary) [58]. The associated rhizobacterial and endobacterial communities associated with *S. alterniflora* were studied via 454 pyrosequencing. The generation of culturable isolates and their biochemical assays showed that these microbial communities displayed potential ecological functions via the production of 1-aminocyclopropane-1-carboxylate (ACC) deaminase, plant growth, and enhanced uptake of HMs [58].

The increased levels of heavy metals in soil are phytotoxic and pass through the plant vascular tissues and interfere with biomolecule (DNA, protein, etc.) functions in the plants. When exposed to prolonged conditions of HM stress, some plants have evolved adaptive mechanisms such as morphological features (namely trichomes, cuticle, hairy roots, etc.) and physiological adaptations (phytohormone production, root exudates secretion, accumulation of proline, etc.) [59,60,61]. The plants show tolerance mechanisms via the uptake of HMs and efflux, sequestration, and transport [62]. Some key examples of metal-tolerant plants comprise *Deschampsia caespitosa*, *Arabidopsis arenosa*, *Silene vulgaris,* and *Arabidopsis helleri* [63]. As per their tolerance mechanisms, plants can be classified as plant species that tolerate HM concentrations and are hyperaccumulators and plant species that do not uptake HM and are non-accumulators of metals. Some hyperaccumulator plant species include *Combretum erythrophyllum*, *Azolla filiculoides,* and *A. halleri*, etc., capable of growing in HM-contaminated soils. These plants deal with HM stress via high metal chelator concentrations, enhanced detoxification and bioaccumulation, and transport system overexpression in the aerial plant tissues [62].

Studies in recent decades have emphasized the microbiome’s role in plant health and responses during stress conditions, among others [57,64]. These microbes are powerful candidates for gaining insights on HM stress alleviation in crop plants and research to understand the function of the microbiome in HM tolerance. Plant-associated endophytes regulate morphological and metabolic processes in plant hosts via multiple detoxification mechanisms, comprising fungal cell wall binding, scavenging, precipitation, extracellular scavenging, vaporization, and compartmentalization, including other mechanisms [65].

## 3. Omics Biology in Deciphering Plant Microbiome and Alleviation of Heavy Metal Stress

In the present decade, plant-associated microbiomes are gaining significant recognition as biological alternatives for heavy metal tolerance and mitigation. The recent advances in high-throughput sequencing technologies have opened new avenues in the characterization of microbiomes, deciphering the functional mechanisms of these microbes. Omics approaches, namely metagenomics, metaproteomics, metatranscriptomics, and metabolomics have proved to be valuable tools to understand microbe composition and structure (diversity, abundance), plant–microbe dynamics, and potential effects on exposure to HM stress [57,66]. Several genes from plants and plant-associated microbes have been identified by employing omics biology and can be further explored for conferring metal tolerance to the holobiont (Table 2).

Photolo et al. [69] performed genome sequencing of an endophyte, *Methylobacterium radiotolerans* MAMP 4754, associated with *Combretum erythrophyllum* (a hyperaccumulator plant), and identified the genes involved in the tolerance of Ni, Cu and Zn metals. Future implications of these studies include the inclusion of information on the metal-tolerant microbiome and the genes in producing initiatives aimed at creating heavy metal-tolerant crops in the future [80].

### 3.1. Metagenomics

In this approach, the sequencing of microbial DNA is carried out directly from the environmental samples, without microbial isolation [81]. Metagenomes are classified as- whole-genome shotgun metagenomics and high-throughput-targeted amplicon sequencing [82]. Shotgun metagenomics delineates the structural and functional characteristics of microbial communities. The elucidation of the ‘HM resistome’ (combination of all the heavy metal resistance genes) of agriculture soil with and without cadmium contamination was determined using shotgun metagenomics [67]. The genes involved in the translocation of HMs were annotated functionally, with P-type ATPases functioning in detoxification and the efflux of cadmium being *czcA*, *czcD*, *czrA*, etc. Moreover, multiple genes involved in Cu, Ni, Fe, and Co resistance, etc., were also identified [67]. Chen et al. [68] studied the bacterial microbiomes of HM-contaminated rivers in China via multiple approaches—16S rRNA gene sequencing, comparative metagenomics, and quantitative PCR studies. The key bacterial species in the core microbiota comprise *Bacteroidetes*, *Proteobacteria*, and *Firmicutes*, showing higher presence. In addition, key insights about genes in DNA repair and recombination and metal-resistant genes in contaminated rivers were also revealed [68]. The studies on microbial metagenomes have also provided key knowledge pertaining to how microbial inoculation in plants improves the phytoremediation capacity of some plants. Fan et al. [83] discussed that *Mesorhizobium loti* HZ76 (rhizobial bacteria) on plant inoculation enhanced the phytoremediation of HM in *Robinia pseudoacacia* in contaminated soil. Further, the shotgun sequencing and 16S rRNA gene sequencing of the rhizospheric microbes showed upregulation of ATP-binding cassette transporter genes, suggesting beneficial attributes of plant–microbe interactions in promoting the efficiency of phytoremediation [83].

On the other hand, high-throughput targeted amplicon sequencing includes the ribosomal RNA genes’ specific amplification (18S rRNA or ITS for fungi and 16S rRNA for bacteria and archaea) and is employed for the determination of diversity and the composition of microbial communities. 16S rRNA gene sequencing is a cost-effective approach and is widely employed for the taxonomic profiling of microbes present in metal-contaminated soil samples. Remenar et al. [84] performed 16S rRNA gene sequencing of Ni-contaminated sites and identified key microbial phyla (*Crenarchaeota* and *Euryarcheota*) present in the region.

### 3.2. Metaproteomics

Additionally, designated as environmental proteomics, metaproteomics includes the high-throughput study of all the proteins present in microbial communities and directly harvested from the environment [85,86]. The steps in the omics approach consist of protein extraction from the environmental samples, protein digestion into peptides, and fractionation employing 2D gel electrophoresis and mass spectrometry-mediated identification of proteins. The fast process aids identification and protein quantification and protein–protein interactions in microbial communities, providing precise insight into the functional roles of microbes. In a key example, Mattarozzi et al. [74] studied the rhizosphere of *Noccaea caerulescens* (a Ni hyperaccumulator plant) and *Biscutella laevigata* (a heavy metal-tolerant plant) inhabiting serpentine soils via LC-HRMS-based metaproteomics and 16S rRNA gene sequencing [74]. The characterization of the microbial communities in the Co-, Cr- and Ni-contaminated soil showed proteins function in response to metal transport and stimulus as well as the presence of key bacterial species—*Stenotrophomonas rhizophila*, *Microbacterium oxidans*, *Bacillus methylotrophicus* and *Pseudomonas oryzihabitans* in the rhizospheric zone [74]. Earlier studies have documented that bacterial genera such as *Streptomyces* and *Pseudomonas* have gradually developed Ni resistance, attributed to the formation of a highly Ni-resistant niche in the soil [87].

### 3.3. Metatranscriptomics

Metatranscriptomics comprises a high-throughput approach used for the detection of active microbes (actively transcribed in the sample) under specific environmental conditions [88,89]. The approach has been quite successful in studying microbial mRNA pool changes in samples and microbial response to heavy metal stress [72]. The gene identification from microbial communities involved in adaptation to harsh environmental conditions has been facilitated by functional metatranscriptomics. The typical steps in metatranscriptomics studies include total RNA isolation from the sample, cDNA library preparation, HM-tolerant transcript screening via yeast or bacterial complementation systems, and transcript sequencing of targeted ones [90,91]. Lehembre et al. [71] employed a functional metatranscriptomics approach for delineating the functional role and mechanisms of microbes in metal resistance. The metatranscriptome library of soil eukaryotes was subjected to functional screening to gain insights into the rescue of Cd- and Zn-sensitive yeast mutants. Some novel proteins such as saccaropine dehydrogenase and BolA proteins (Zn-tolerance) as well as the C-terminal of aldehyde dehydrogenase (ADH) (Cd tolerance) were identified [92]. The enzyme aldehyde dehydrogenase removes the aldehydes (toxic) formed during multiple abiotic stresses.

### 3.4. Metabolomics

Metabolomics is defined as the large-scale study of all low-molecular-weight compounds (<2 kDa) present in a specific environmental condition [93]. Metabolome defines the final stage in omics, best representing the organism’s phenotype. Metabolomics studies have been successful in filling the gaps between the genotype and the phenotype of an organism [64]. Metallophytes are plants that have advanced mechanisms to tolerate high HM concentrations and may be obligate metallophytes, surviving only in high metal concentrations or facultative metallophytes and present in both HM-contaminated and normal sites. In addition, the rhizospheric microbes may improve metal tolerance and assist in phytoremediation [63,94]. The beneficial characteristics of metal-resistant microbes include phytohormone secretion, solubilization of nutrients, and ACC deaminase production, comprising key mechanisms of plant growth promotion in conditions of nutrient deficiency in soil [95,96].

Metabolomics research on metal-tolerant plants and metal-contaminated soils has been remarkable in the identification of metal-tolerant microbes. Another study suggested that plant metabolite production of amino acids, phenols, and organic acids is a process that acts similarly to nutrients for microbes, promoting the phytoremediation of benzopyrene and pyrene as compared to the control [97]. Furthermore, metabolomics studies of 73 metabolomes associated with a wetland grass, *P. australis*, revealed different root areas (rhizosphere vs. endosphere) and secretion of particular metabolites due to the presence of dissolved solutes and different metals and pH [77]. Niu et al. [78] aimed to decipher the HM-tolerance mechanism via metabolomic studies of plants inoculated with or without microbes. The capability of *S. integra* for Pb bioaccumulation was studied with rhizospheric microbe inoculation via targeted metabolomics [78]. In a similar study, increased proline levels and antioxidant activity (increased catalase and superoxide dismutase levels) were observed when the plant was inoculated with Pb-resistant *Aspergillus niger* and *Bacillus* sp. [78]. Han et al. [79] employed a combination of proteomic and metabolomic approaches to understand and elucidate the increased tolerance of *Triticum aestivum* to Pb and Cd stress upon *Enterobacter bugandensis* TJ6 strain inoculation via multiple mechanisms, including extracellular absorption and increased bio precipitation, reduced bioaccumulation of Cd and uptake of Pb and IAA, arginine and betaine secretion, enhancing metal tolerance and mitigation in the stress condition [79].

## 4. Harnessing Plant Endophytic Microbiome for Heavy Metal Tolerance and Mitigation

Plant microbiomes are constantly influenced by any changes in the immediate environment, with the rhizosphere region comprising the ‘hot spot’ for plant–microbe interactions governed by different root exudates [98]. Studies have shown improved stress tolerance by plants, with alterations in the root microbiome [11]. The presence of HMs, namely Co, Cu, Zn, and Mn, is natural and essential for the smooth functioning of key biological processes. Plant growth and characteristics are adversely impacted by alterations, demonstrated by decreased photosynthesis, chlorosis, restricted growth, low biomass, alterations in water balance, and senescence, leading to plant death [99]. The existence of HMs in soil induces signaling pathways, including hormone signaling pathways and ROS pathways, enhancing the expression of genes in stress response [100].

While multiple bacterial endophyte strains have been extensively explored and documented as efficient alternatives in increased stress tolerance and plant growth promotion, recent times have witnessed research undertaken on fungal endophytes as emerging candidates in phytoremediation [17]. The utilization of plant endophytic microbiome comprises a prospective approach in bio-based phytoremediation aimed at restoring environmental health and sustainability. Bacterial endophytes colonize internal plant tissues and complete their life cycle without causing any adverse effects on the plant hosts. Bacterial endophytes form small aggregates in plant tissues to aid nutrient exchange, increase microbial metabolism, and lead to the physiological functions of the host plant [101]. Endophytes promote plant growth in multiple ways—biological nitrogen fixation, phytohormone synthesis, siderophore production (metal-chelating substances), and solubilization of minerals [102]. Furthermore, bacterial endophytes are well adapted to HM-contaminated sites and display resistance mechanisms via metal bioaccumulation and enzymatic oxidation/reduction of metals to non-toxic forms [23]. Figure 2 provides a schematic representation of molecular mechanisms adopted by bacterial endophytes in the tolerance and mitigation of HMs. Bacterial endophyte promotes HM mitigation via multiple detoxification mechanisms, intracellular bioaccumulation of metals, and metal immobilization via the production of siderophores, organic acids, and biosurfactants and shows alterations in the activity of metal-induced antioxidant enzymes.

Additionally, certain microbes display mechanisms comprising stress-related gene expression and an increase in metal accumulation in plants [103]. The production of ACC deaminase by multiple bacterial species influences metal tolerance to vary degrees via alterations in plant ethylene levels [104]. Studies have documented the role of dark septate fungi (DSE) as efficient promoters of multiple stress tolerance [105]. Rho et al. [106] researched the response of 94 endophyte strains in 42 plant species to abiotic stresses (nitrogen deficiency, salinity, and drought) and their effects on plant overall fitness [106]. The endophyte strains demonstrated different levels of stress mitigation effects and increased plant biomass under all three stresses. The results showed the efficacy of endophytes in the mitigation of salinity, nitrogen deficiency, and drought in a different range of plant host species.

To understand whether bacterial endophytes are an ideal candidate for HM tolerance (as suggested by Rho et al. [106]), Franco-Franklin and coworkers [26] performed a meta-analysis of the literature on bacterial endophyte and HM tolerance from the last 10 years and estimated the effective microbial size. The study concluded an overall positive effect on plant growth and metal tolerance in different species, suggesting the beneficial prospects of endophytes as biological agents in HM tolerance and mitigation [26].

### 4.1. Characterization of Plant-Associated Endophytes and Their Dynamics

In recent times, the importance of microbiomes in the modulation of plant defense response has been increasingly recognized and substantially contributes to gaining knowledge/insights on plant–microbe dynamics and their implications in growth and survival. Exposure to abiotic stressors including nutrient deficiency [107], drought [108] and metal toxicity [4], etc., alters metabolism in plant roots and microbial communities.

Recently, the cointegration of metagenomics with synthetic communities and metabolomics has been significant in addressing the hurdles in research on plant–microbe interactions [109,110]. In addition, the application of synthetic microbial communities in plants facilitates the understanding of the biological significance of such changes [11]. Another prospective approach for defining the biochemical diversity of root exudates (for presence or changes in metabolites concentration) is to understand the root exudates-driven microbiome assembly during stress conditions. In the perspective review, Liu et al. [7] shared a hypothesis on the mechanisms of commensal microbes in the mitigation of plant antagonisms, including (i) compound production to mitigate adverse impact, e.g., enzymes to scavenge ROS generation, (ii) alterations or degradation of microbe-associated molecular patterns (MAMPs) and thereby decreased plant response [111] and (iii) the presence of cell surface molecules that induce plant defenses [112]. The recent advances in high-throughput technologies and omics biology have substantially improved our understanding of how these plant-beneficial microbes offer prospective solutions to tackle the high concentration of HM in the soil. Some key studies in this area have witnessed translational success; however, multiple challenges still exist in terms of long-term subsistence, a feasible genetic chassis, and host metabolite exploration, including others [113].

In the present decade, researchers have extensively explored the potential of plant-associated endophytes in the alleviation of multiple stresses—not exclusively HM stress. Endophytes comprise the microbes that colonize the plant’s internal tissues without causing any adverse changes in the plant [114,115]. De Bary et al. [116] coined the term ‘endophytes’ to collectively designate all the microbes that live inside healthy plants. Wilson [117] further suggested that bacteria and fungi are collectively regarded as endophytes, causing asymptomatic infections within plant tissues and complete partial or total life cycles within host plants. As per the pathogenicity, these categories of endophytes were proposed: (i) non-pathogenic, (ii) pathogenic in another host and nonpathogenic as an endophyte, and (iii) pathogens that have become nonpathogenic but may colonize by selection [118]. Except for some seed-transmitted endophytes, the colonization of endophytes in the plant hosts proceeds through various stages, such as host identification, recognition, plant surface colonization, and entry into plant tissues [119]. The fitness achieved by each partner (microbe and plant host) is largely attributed to the positive implications of plant-endophyte symbiosis. Plant-associated endophytes have immense importance in promoting plant growth (deterrence of pathogens and herbivory), efficient water usage (stomatal regulation, etc.), uptake of nutrients (changes in nitrogen metabolism and accumulation), and countering environment-induced stresses. In return, endophytes are transferred to the next generation and acquire the plant’s nutrients [119]. The coevolution of plants with their associated microbial counterparts has benefitted plant adaptation/survival in challenging abiotic and biotic stress conditions (salinity, pathogen attack, drought, and toxic pollutants/contaminants), leading to ecological subsistence [17,120].

### 4.2. Endophytes and Molecular Mechanisms in Heavy Metal Tolerance

To circumvent metal stress, bacterial endophytes have evolved multiple mechanisms, such as metal ions efflux outside of the cell, conversion of metal ions to less-toxic forms, precipitation, and biomethylation of metal ions [28,121]. Recent studies have highlighted that inoculation of plant seeds/soil with bacterial endophyte (metal resistant) promoted plant growth and enhanced phytoremediation in metal-contaminated soil via improved nutrient uptake, cell elongation, phytostabilization, and metal stress alleviation [122,123,124]. In the present time, fungal endophytes are also emerging as prospective candidates in HM tolerance and mitigation. Upcoming trends in achieving bio-based remediation of heavy metals in contaminated sites via employing endophytic microbes have witnessed translational success. Plant-associated endophytes facilitate their plant hosts to survive and adapt in harsh conditions via pollutant mobilization, plant growth, and increased tolerance to abiotic and biotic stresses [125]. Multiple stress-countering mechanisms are adopted by plant-associated endophytes.

#### 4.2.1. Direct and Indirect Promotion of Plant Growth

The plant growth-promoting microbes assist in the growth of their plant hosts and the mechanisms include the solubilization of minerals, nitrogen fixation, phytohormone production, and production of siderophores, etc., aiding in plant development and metal toxicity mitigation via one or more of these mechanisms [126,127]. Bacterial endophytes possessing nitrogen-fixing properties support plant growth in a nitrogen-deficient environment and enhance plant growth [128]. Doty et al. [129] studied the bacterial endophytes, *Rahnella*, *Acinetobacter*, *Burkholderia,* and *Sphingomonas* (isolated from *Salix sitchensis* and *Populus trichocarpa*), which and improved nitrogen availability in the plants. Another key study by Nautiyal et al. [130] demonstrated that metal-resistant bacterial endophytes solubilized precipitated phosphates in the soil through ion exchange, chelation, and acidification, etc., and promoted phosphorous mineralization via acid phosphatase secretion [131]. The endophyte species, e.g., *Pseudomonas*, *Azotobacter*, *Staphylococcus*, *Azospirillum*, etc., produce phytohormones [132] and are beneficial for plant growth. Some studies have shown bacterial endophytes carry out ACC hydrolysis and reduce ethylene production, thereby alleviating stress [133]. *Trichoderma* H8 and rhizospheric strain *Aspergillus* G16 promote *B. juncea* (L.) Coss. var. foliosa growth in metal-contaminated soils [134]. Fungal endophyte *Lasiodiplodia* sp. MXSF31 (resistant to Pb, Cd, Zn), isolated from *Portulaca oleracea*, increased *B. napus* L. biomass [135].

The indirect mechanisms demonstrated by endophytes deal with decreasing the stress-related effects in plants via induced systemic resistance (ISR) or the biological control of plant pathogens [136].

**Figure 2 plants-12-01515-f002:**
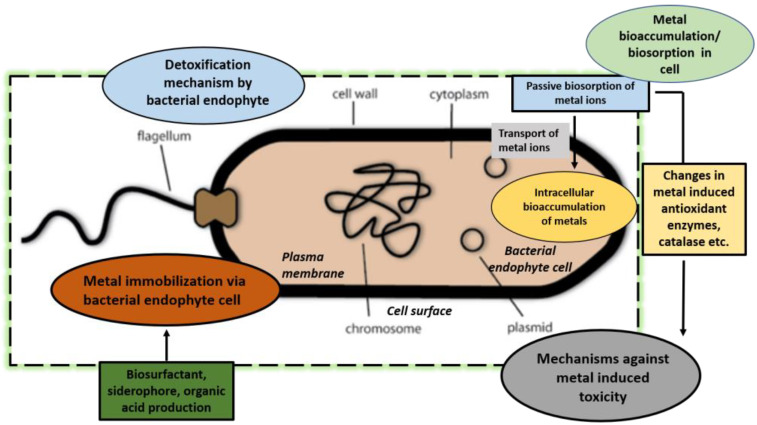
Provides a schematic representation of molecular mechanisms adopted by bacterial endophytes in the tolerance and mitigation of HMs. Bacterial endophyte promotes HM mitigation via multiple detoxification mechanisms, intracellular bioaccumulation of metals, and metal immobilization via the production of siderophores, organic acids, and biosurfactants and shows alterations in the activity of metal-induced antioxidant enzymes (recreated from Ma et al. [136]).

Some endophyte species restrict plant diseases through siderophore production, antibiotics, antimicrobials, and hydrolytic enzymes [137]; for example, bacterial endophytes, *Curtobacterium luteum* TC10 and *Bacillus megaterium* BP17 restrict *Radopholus similis* Thorne (nematode) via antibiotic synthesis [138]. Kloepper and Beauchamp [139] discussed that the bacterial endophyte induces ISR as a defense mechanism against multiple bacterial, viral, and fungal pathogens. Moreover, the plant defense system is induced by priming plants with inoculation of bacterial endophytes and needs to be overcome for host colonization [140]. Once the defense genes are expressed, multiple ISR-induced defense responses, namely the increased function of β-1,3-glucanases, guaiacol, peroxidases, chitinases, and superoxide dismutase [141], allow these enzymes to protect bacterial cells against oxidative stress [47]. Sharma et al. (2022) studied the effect of fungal seed endophyte *Dysphania ambrosioides* on the accumulation and tolerance of Cd and Zn metals in the plant, defining key prospects in decreasing metal toxicity and promoting metal phytoextraction in important agriculture crops [142].

#### 4.2.2. Plant Metal Uptake and Metal Stress Alleviation

The process of phytoremediation is adversely impacted by the occurrence of metal phytotoxicity [143]. Several mechanisms are undertaken by the bacterial endophytes such as the transformation of toxic metal ions to less-toxic forms [123], bioaccumulation and sequestration in intracellular compartments [143], extracellular precipitation [144] and metal ions adsorption/desorption, respectively [145]. The phenomenon of horizontal gene transfer (HGT) in microbes is of value in conferring antibiotic resistance or HM resistance [146]. Zhang et al. [147] showed that bacterial endophytes alter antioxidant enzymes (function in plants glutathione peroxidase, CAT, SOD, ascorbate peroxidase, etc.) and lipid peroxidation opposes HM-induced plant oxidative stress. Moreover, the effect of fungal endophyte was evaluated on plants’ different developmental stages and improvement of Cd tolerance was studied [148]. The study suggested that plant inoculation with fungal endophyte triggered multiple protective mechanisms to counter Cd stress [148].

In some cases, the mechanism of methylation by the bacterial endophyte is also suggested; for instance, the bacterial endophytes (Hg-resistant) express *merA* gene coding for mercuric reductase convert highly toxic ionic Hg^2+^ into less toxic volatile Hg^0^ [149], and *merB* gene coding for organomercurial lyase converts organomercurials into mercuric ion (Hg^2+^) [150], improving phytovolatilization. However, further knowledge is required on whether plants inhibiting metal-contaminated soils affect the survival and colonization of beneficial metal-resistant microbes.

#### 4.2.3. Endophytes Enhance Metal Bioavailability

The metal bioavailability in the soil is a primary factor that affects HM transfer from the soil to the plant [151]. Research has documented that the bacterial endophytes comprise a metal resistance/sequestration pathway (e.g., *ncc-nre*) that plays a predominant role in pollutant phytotoxicity alleviation [152] and improves HM phytoavailability via soil acidification, the solubilization of phosphates, production of chelating agents and redox activity [153]. For example- the bacterial endophyte *Bacillus cereus* enhanced Cd uptake and promoted phytoremediation when inoculated in *Phytolacca acinosa* [154]. In addition, the endophyte increased the bioavailability of P and Cd in the rhizosphere and promoted plant growth [154]. In addition, bacterial endophytes are capable of biosurfactant production and release as root exudates that interact and form complexes with insoluble metals, performing metal desorption and resulting in metal movement and increased bioavailability in the soil [126]. Moreover, the bacterial endophyte produces extracellular polymeric substances (EPS) (comprising proteins, nucleic acids, polysaccharides, and lipids) that form metal complexes, reducing their bioaccessibility [121].

#### 4.2.4. Uptake of Metals and Translocation by Endophytes

Endophytes affect the metal translocation and bioavailability in plants, altering toxicity via multiple metabolite secretions including organic acids, siderophores, etc. [155,156]. In a key study, Sun and coworkers [157] discussed that plant inoculation with bacterial endophyte (showing plant growth promotion) promoted the transfer of Cu metal from the root to above-ground tissue of *B. napus* and led to improved phytoextraction [157]. Ma et al. [153] showed that *Pseudomonas* sp. A3R3 (Ni-resistant bacterial endophyte) efficiently increased plant biomass (*B. juncea*) and Ni accumulation (*A. serpyllifolium*) when grown in Ni-contaminated soil. These results can be attributed to the secretion of growth-promoting substances (IAA, ACC deaminase, etc.) and polymer hydrolyzing enzymes (cellulase, etc.), highlighting the positive impact of endophytes in the phytoremediation of metal-contaminated soils.

#### 4.2.5. Bioaccumulation of Metals and Biosorption

Bioaccumulation of metals and their biosorption defines the crucial process in endophyte-mediated bioremediation. In another study, *Serratia* sp. LRE07, a Cd-resistant bacterial endophyte, efficiently absorbed Zn and Cd in bacterial cells, thereby decreasing metal phytotoxicity [158]. The process of the endophyte-assisted biosorption of metals consists of bioaccumulation/active absorption, metal uptake (cell transport, intracellular accumulation, transport, etc.) via active metabolism-dependent transport into cells [159], and passive biosorption by dead and living cells occurring in the cell wall via metabolism-independent processes [160]. In this direction, studies have suggested various metal-binding mechanisms, namely complexation, coordination, ion exchange, and microprecipitation, among others that show synergistic involvement [161]. The remarkable capacity of endophytic strains for metal bioaccumulation greatly enhances bio-based HM detoxification and promotes phytoremediation [143].

### 4.3. Genetic Engineering-Mediated Chassis of Endophytes and Enhanced Tolerance to Heavy Metals

The present era has witnessed increased exploration and investigations on endophytes, attributed to their emerging popularity in socio-economic contexts. However, to date, whole-genome sequencing has been undertaken for only a few species, such as fungal endophyte genomes including *Sarocladium brachiariae* [162], *Laburnicola rhizohalophila* [163], etc., and bacterial endophyte genomes including *Pseudomonas putida* W619, *Enterobacter* sp.638, *Methylobacterium populi* BJ001, *Serratia proteamaculans* 568, etc. A study is underway at the United States Department of Energy Joint Genome Institute (http://www.jgi.doe.gov (accessed on 20 February 2023)), providing deep insights into the plant–endophyte dynamics and molecular mechanisms of endophytes.

In addition to multi-faceted applications, multiple endophyte strains have demonstrated inherent capacity for the degradation of environmental pollutants/xenobiotics or for acting as vectors to introduce degradation traits into the desired organisms [164]. In addition, the multiple attributes of endophytes in the degradation of organic compounds and HM resistance probably originate from their prior exposure to different compounds present in soil/plant niches [28]. As early as 2001, Siciliano and coworkers [165] demonstrated that plants growing in xenobiotic-contaminated soil are rescued by endophytes attributed to the presence of pollutant-degrading genes in the endophyte genomes [165]. The study further discussed that endophytes associated with the plants inhabiting the nitro-aromatic-contaminated soil showed more prevalence of genes for the degradation of the nitro-aromatic compound. Van Aken and coworkers [166] discussed that the endophyte *Methylobacterium*, isolated from *Populus deltoids x nigra* (a hybrid poplar), efficiently degraded nitro-aromatic compounds such as 2,4,6-trinitrotoluene, highlighting that to confer or improve biodegradation capacity, an engineered endophyte strain may be a prospective candidate for HM phytoremediation. The genetically manipulated strain of *Burkholderia cepacia* G4 conferred its plant host, showed tolerance to toluene, decreased transpiration in the environment, and the potential of endophyte-assisted phytoremediation via reducing cytotoxicity and efficient xenobiotic degradation [167]. Newman and Reynolds [168] studied the beneficial attributes of endophytes to improve xenobiotic remediation, suggesting that the genetic engineering of bacterial endophytes is a feasible approach as microbes can be easily manipulated compared to their plant counterparts.

The genetic manipulation of an endophyte strain necessitates pathway engineering in the desired organism (xenobiotic degradation pathway) and expression of pollutant degrading genes for the assessment of efficient bioremediation. The plant hosts interior environment provides the endophyte strain (possessing degradation properties) access to large population sizes, owing to less competition. Another key advantage of employing a degradative endophyte strain is that any pollutant is degraded in planta via uptake, limiting the negligible effects on the surrounding herbivorous fauna. Studies have also documented that bacterial endophytes that express the necessary degradative genes may increase xenobiotic degradation or their analogs via translocation or bioaccumulation processes. Several plant growths promoting bacteria from the genus *Pseudomonas*, *Streptomyces*, *Bacillus,* and *Methylobacterium* have been isolated with the potential of HM-tolerance and the mitigation of adverse effects of HMs on plant hosts [96]. Recently, Tang and coworkers [169] studied the prospects of endophytes, *Phomopsis columnaris,* and *Setophoma terrestris* isolated from *Dysphania ambrosioides* in bioaccumulation under multiple HM stresses and metal tolerances. The results showed that the beneficial effects of endophyte inoculation improved plant growth and HM tolerance via antioxidant mechanisms. In addition, multi-isolate inoculation in the plant host had better effects than a single isolate on the host plant [169]. Further studies by Bibi et al. [170] discussed the efficient biodegradation of Cr metal in the contaminated soil via fungal endophytes, and 114 fungal endophyte strains were isolated from different plant species, namely *Zea mays*, *Beta vulgare*, *Saccharum officinarum*, *Ficus carica*, *Parthenium argentatum*, *Cucurbita maxima,* and *Colocasia esculenta*. The isolated fungal endophyte strains possessed the ability to mitigate Cr toxicity in *Lactuca sativa* L. and promote plant growth and metal tolerance in metal-contaminated soil [170]. In another study, the potential impact of endophyte *Mucor* sp. MHR-7 in the alleviation of HM contamination was studied [171]. The results showed that the endophyte *Mucor* sp. MHR-7 was capable of accumulating up to 94% HM in its hyphae and acting as an efficient phytostimulant via the secretion of ACC deaminase, IAA, and phosphate solubilization [171].

With advances in synthetic biology and whole genome sequencing, the genetic manipulation of beneficial microbes in a socio-economic context is rapidly gaining momentum. The pollutant-degrading genes from biological organisms (microbes, plants) can be introduced into another microbe or plant [172], and the heterologous expression in the transgenic organism may improve phytoremediation efficiency, increase organic pollutants removal and raise tolerance, compared to the natural counterparts [173,174]. The genetic manipulation of endophyte strains for degradation pathway reconstitution defines an attractive approach to efficient pollutant degradation in plant vascular systems or rhizospheres [175]. The recombinant bacteria engineered with degradation pathways and their subsequent plant inoculation facilitate microbe colonization and phytoprotection against naphthalene and increase degradation [55]. In another prospective study, yellow lupine and poplar, when inoculated with *Burkholderia cepacia* VM1468, engineered endophytes, promoted plant biomass and reduced the evapotranspiration and phytotoxicity of TCE and toluene [176,177]. In addition, the engineered microbial strain may act as a vector for the transfer of the desired traits (degradation properties in the present context) for the improved efficiency of organic pollutant remediation, either via transfer in the endogenous community via microbial conjugation or the transfer of the degradation genes via HGT among endophyte strains. The pTOM-Bu61 plasmid (coding for combined TCE and toluene degradation) was transferred among bacterial endophytes in plants via the phenomenon of HGT [176], defining beneficial phenomena in phytoremediation and environmental subsistence. The natural process of HGT plays a prominent role in the evolution of endophytes that are naturally engineered for the presence of degradative genes and heterologous expression, facilitating rapid adaptations of new endophyte communities to stress conditions [176]. In plants, the presence of degradative plasmids and *nah* (naphthalene dioxygenase) genes suggests that HGT phenomena can assist in the spread of *nah* genes, benefitting adaptation of endophyte and the degradation of petroleum hydrocarbon in planta [178].

The genetic engineering of prospective endophyte candidates via synthetic biology defines novel paradigms in the creation of transgenic strains with novel attributes. An interesting discussion by Chowdhary et al. [179] highlights the prospects of CRISPR/Cas9-based genome editing in endophytes as a novel strategy to enhance secondary metabolite production in endophytes. The elucidation of endophyte genomes has provided valuable insights into the genome, facilitating the tagging of biosynthetic gene clusters (BGCs) and precise detection rapidly, with the CRISPR/Cas9 tool efficiently introducing multiple site mutations in a single procedure [179]. Wen-Jia and coworkers [180] directed CRISPR/Cas9-mediated gene cluster-knockout system in *Streptomyces* SAT1 (a secondary metabolite gene cluster) for cluster gene editing for secondary metabolite production and the inhibition of chestnut blight disease, highlighting the antimicrobial potential of endophytes [180]. Furthermore, genome editing approaches (employing CRISPR/Cas9 system) as prospective approaches to induce/enhance the production of novel metabolites have been discussed in relation to filamentous fungi [181].

## 5. Perspectives and Future Outlook

Substantial efforts are being made to promote plant health and productivity in challenging conditions of abiotic stress, but limited success has been achieved, attributed to the limited knowledge in understanding stress tolerance mechanisms. Plant microbiomes play a crucial role in conferring stress tolerance in plants and producing better adaptability and survival in fluctuating environmental conditions. In recent times, plant-associated endophytes have gained considerable recognition in the promotion of plant growth and productivity, abiotic/biotic stress tolerance, and as biocontrol agents, highlighting the need for further exploration of plant endophytic microbiome and its dynamics within plant hosts. Although several studies have documented and highlighted the stress alleviation potential of endophytes, investigations on their prospects in HM phytoremediation are increasingly recognized only in the present era. The inherent capacity of plant-associated endophytes in the tolerance and mitigation of HMs can be enhanced via employing synthetic biology-mediated approaches, defined by the successful attempts in CRISPR/Cas9-based genome editing for enhancing valuable socio-economic traits in engineered endophytes. However, knowledge gaps in the genetic chassis of endophytes still need to be addressed and bridged, and increased investigations on plant endophytic microbiomes are essential for efficient utilization in crop productivity and ecological subsistence.

## Figures and Tables

**Figure 1 plants-12-01515-f001:**
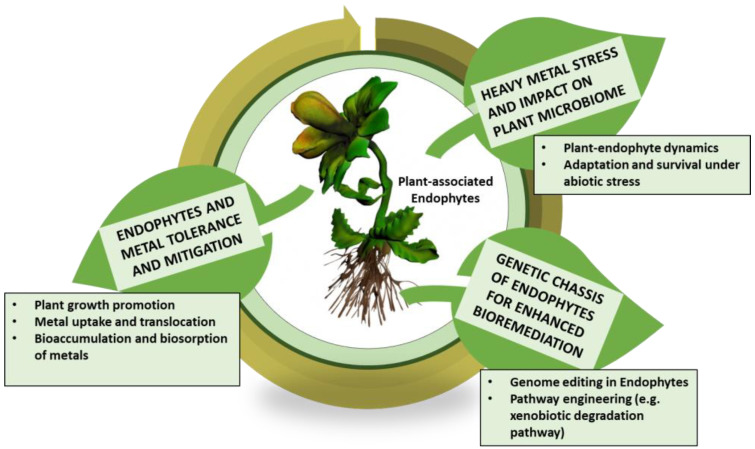
Recent trends in harnessing plant endophytic microbiome for HM mitigation in plants. Efforts have been made to understand and discuss the implications of HMs on plant–endophytic dynamics and the beneficial effects of endophytes on the phytoremediation of HMs and tolerance. Furthermore, the genetic chassis of endophytes (via CRISPR-Cas-based genome editing and pathway engineering) can create high-value strains with potent efficacy towards tolerance and enhanced HM alleviation from contaminated sites.

**Table 1 plants-12-01515-t001:** Discussion of the bioremediation potential of plant-associated endophytes in the tolerance/mitigation of HM-associated stress in plants, highlighting key examples.

Endophyte Strain	Plant Host	Pollutants/Contaminants	Mechanisms	Reference
*Enterobacter* sp. CBSB1 (genetically engineered strain)	*Brassica juncea*	Cd and Pb	Metal-tolerance and phytoremediation	[29]
Endophyte consortia	*Agrostis* *stolonifera*	Pb	Higher Pb tolerance, plant growth	[30]
*Bacillus pumilus* E2S2	*Sedum* *plumbizincicola*	HM	HM phytoextraction from the soil	[31]
*Rahnella* sp. JN6	*Polygonum* *pubescens*	Cd, Pb, and Zn	Metal-tolerance and alleviation	[32]
*Methylobacterium* *oryzae* CBMB20, *Burkholderia* sp.	*Lycopersicon* *esculentum*	Ni, Cd	Biosorption of Cd and Ni, Phytohormone synthesis, ACC deaminase activity	[33]
Bacterial endophyte *PRE01*	*Pteris vittata*	V, Cd, Cr	Production of siderophore, ACC deaminase, indoleacetic acid (IAA), HM detoxification, metal uptake	[34]
*Pseudomonas azotoformans* ASS1	*Trifolium* *arvense*	Zn, Ni, Cu	Phytoremediation of Ni, Zn, Cu	[35]
*Bacillus, Stenotrophomonas, Enterobacter*	*Pteris vittata*	As	Plant growth promotion, high As tolerance, phytoextraction of metal	[36]
Bacterial endophytes	*P. vittata*	As	High As tolerance, IAA production enhanced phytoremediation	[37]
*Serratia* PRE01, *Arthrobacter* PRE05	*B. juncea*	V	Improved endosphere and rhizosphere micrology, enhanced phytoremediation	[38]
*Pantoea stewartii* ASI11, *Enterobacter* sp. HU38, *Microbacterium arborescens* HU33	*Brachiaria* *mutica*, *Leptochloa fusca*	Cr	Plant growth promotion, increased plant biomass, Cr uptake and translocation, phytostabilization of Cr	[39]
*Microbacterium arborescens* HU33, *Pantoea stewartii* ASI11	*L. fusca* (L.) Kunth	Pb, U	Plant growth, enhanced phytoremediation, phytostabilization of U- and Pb-contaminated soils	[40]
*Bacillus amyloliquefaciens* RWL-1	*Oryza sativa*	Cu	Plant growth, reduces the metal accumulation	[41]
*Microbacterium* sp. G16, *Pseudomonas* *fluorescens* G10	*Brassica napus*	Pb	Phytoextraction of Pb	[42]
*Pseudomonas* sp., *Microbacterium* sp.	*Rumex acetosa*	Mixed HM	Enhanced phytoremediation of HM	[43]
Bacterial endophyte consortia	*Lupinus luteus*	Metals and organic pollutants	Phytoremediation of metals and organic pollutants in contaminated site	[44]
*Penicillium* sp. CBRF65, *Fusarium* sp. CBRF44, *Alternaria* sp. CBSF68	*B. napus*	Cd and Pb	Detoxification and remediation of HM	[45]
*Bacillus thuringiensis* GDB-1	*Pinus sylvestris*	HM	Enhanced remediation of HM	[46]
*Serratia nematodiphila* LRE07	*S. nigrum*	Cd	Plant growth, phytoremediation of Cd	[47]
*Microbacterium* *lactium* YJ7	*B. napus*	Cu	Cu uptake and phytoextraction	[48]

**Table 2 plants-12-01515-t002:** Provides insights into the key genes, proteins, and metabolites involved in heavy metal tolerance discovered via omics approaches/high-throughput technologies.

Techniques Used	Biological Resource/Sample	Key Genes, Proteins and Metabolites and Their Function	Reference
Metagenomics	Microbial communities in agricultural soil	*czcA*, *czcD* and *czrA* in detoxification and efflux of cadmium	[67]
16S rRNA gene sequencing, comparative metagenomics, and quantitative PCR	Bacterial microbiomes	Identification of metal-resistant genes	[68]
Genome sequencing	*Methylobacterium radiotolerans* MAMP 4754 (endophyte)	Genes involved in the tolerance of Ni, Cu, and Zn metals were identified	[69]
Metagenomics	Soil	Genes encoding for ABC transporters, detoxification process	[70]
Functional metatranscriptomics	Soil microbiota	Novel proteins, BolA proteins, saccaropine dehydrogenase (for Zn tolerance), and the C-terminal of aldehyde dehydrogenase (ADH) for Cd tolerance were identified	[71]
Metagenomics, metatranscriptomics	Soil microbiota	Genes involved in Cr resistance, transport, and reduction	[72]
Metagenomics, metatranscriptomics	Cr-contaminated soil	Identification of six novel genes having Cr tolerance (including *gsr* and *mcr*)	[73]
LC-HRMS-based metaproteomics and 16S rRNA gene sequencing	*Noccaea caerulescens* (Ni hyperaccumulator), *Biscutella laevigata* (HM-tolerant)	Proteins involved in metal transport (Ni, Cr, Co) and response to stimuli were identified	[74]
Proteomics	*Arabidopsis halleri*	Upregulation of stress-related proteins (superoxide dismutase, rubisco, and malate dehydrogenase) and decrease in defense-related proteins	[75]
Metabolomics, Transcriptomics	*Sedum alfredii*	Higher Cd phytoremediation via lateral root formation.	[76]
Metabolomics	*P. australis*	Spatial metabolite secretion due to different solutes, pH, and presence of different metals	[77]
Metabolomics	*Salix integra*	Identification of 401 metabolites (carbohydrates, organic acids, and amino acids)	[78]
Proteomics and metabolomics	*Enterobacter* *bugandensis* TJ6	Betaine, arginine, and IAA secretion	[79]

## Data Availability

The data reported in the study can be found within the article.

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
