# Peer review of "Trends in Harnessing Plant Endophytic Microbiome for Heavy Metal Mitigation in Plants: A Perspective"

_plants, 2023, doi:10.3390/plants12071515_

Round 1

Reviewer 1 Report

1.     The English grammar in this paper must be improved for the reconsideration of publication. The manuscript needs to be edited for grammar and syntax. For example, Line 28, “plant growth and yield 50 and 70%”?

2.     Line 109. Figure 1. Recent trends in Harnessing Plant Endophytic Microbiome for??

3.     Line 118. Pls revise the figure caption and add more details in figure caption for the information incorporated in the figure.

4.     Line 180. czcA, czcD, czrA, the gene names should be italic, pls check the similar mistakes throughout the manuscript.

5.     Line 279. You used the abbreviated names of some typical metals, but here you used cobalt, copper, zinc, and manganese, pls revise the description. I also found this mistake in other part of the manuscript.

6.     Line 283 & 289. As for ROS, you can use the full name with its abbreviated name in the bracket in first use then, use the abbreviated name in the following parts.

7.     Figure 2 is not clear enough and the figure caption needs to be detailedly-described for the important information in the figure as I mentioned in Q3.

8.     Sections 3 and 4 share some overlap contents, pls check and combine them if possible.

Author Response

Reviewer 1

  1. The English grammar in this paper must be improved for the reconsideration of publication. The manuscript needs to be edited for grammar and syntax. For example, Line 28, “plant growth and yield 50 and 70%”?

Response-The authors thank the reviewers for the suggestions/comments made for the overall improvement of the manuscript. The manuscript was extensively revised for English grammer and syntax for improving the clarity and consistency, please see the revised manuscript for changes. Line 28, “plant growth and yield 50 and 70% was corrected as Besides biotic stress, abiotic stresses adversely impact plant physiology, functions and productivity, affecting plant growth and yield upto 50 and 70%, in most plant species.

  1. Line 109. Figure 1. Recent trends in Harnessing Plant Endophytic Microbiome for??

Response- The sentence was corrected, Figure 1. Recent trends in harnessing plant endophytic microbiome for HM mitigation in plants. Please refer to the revised manuscript.

  1. Line 118. Pls revise the figure caption and add more details in figure caption for the information incorporated in the figure.

Response-The figure captions for figure 1(line 118) was revised and more details was added, please see the revised caption.

Figure 1. Recent trends in harnessing plant endophytic microbiome for HM mitigation in plants. Efforts have been made to understand and discuss the implications of HM on plant-endophytic dynamics and the beneficial effects of endophytes on the phytoremediation of HM and tolerance. Furthermore, the genetic chassis of endophytes (via CRISPR-Cas-based genome editing and pathway engineering) can create high-value strains with potent efficacy towards tolerance and enhanced HMs alleviation from con-taminated sites.

  1. Line 180. czcA, czcD, czrA, the gene names should be italic, pls check the similar mistakes throughout the manuscript.

Response-We agree by the suggestions indeed the gene names should be in italics and were revised accordingly, similar mistakes were checked and corrected in the revised manuscript.

  1. Line 279. You used the abbreviated names of some typical metals, but here you used cobalt, copper, zinc, and manganese, pls revise the description. I also found this mistake in other part of the manuscript.

Response-The reviewers are thanked for the comment. We have corrected the mistake and now all the metals are denoted with abbreviation in the revised manuscript, please refer to the manuscript for changes.

  1. Line 283 & 289. As for ROS, you can use the full name with its abbreviated name in the bracket in first use then, use the abbreviated name in the following parts.

      Response- In line 283, 289, at first mention, full name of Reactive oxygen species (ROS) was used, subsequently, abbreviation ROS was used, please see the changes made.

  1. Figure 2 is not clear enough and the figure caption needs to be detailedly-described for the important information in the figure as I mentioned in Q3.

Response-The clarity of figure 2 was low. As per the suggestions, the figure 2 was redrawn and revised for clarity and consistency and a detailed caption was added for better representation of the information in the figure. Please see the revised manuscript for changes.

Figure 2. Provides a schematic representation of molecular mechanisms adopted by bacterial endophytes in the tolerance and mitigation of HMs. Bacterial endophyte promotes HM mitigation via multiple detoxification mechanisms, intracellular bioaccumulation of metals, metal immobilization via the production of siderophores, organic acids, and bio-surfactants, and alterations in the activity of metal-induced antioxidant enzymes (recreated from Ma et al. [136]).

  1. Sections 3 and 4 share some overlap contents, pls check and combine them if possible.

Response-The section 3 and 4 shared literature overlaps. The authors have tried their best to remove the similar portions. However, we did not combine the two sections 3 and 4, because these are different and discuss the different aspects of heavy metal tolerance.

Reviewer 2 Report

The manuscript, ‘Trends in Harnessing Plant Endophytic Microbiome for heavy 2 metal mitigation in plants: A perspective’ is well written, compiled, and informative. It has included crucial and contemporary aspects of Heavy metal tolerance in the plants conferred by endophytes. The manuscript is worth publishing however a few minor comments are being made, author may consider including:

  • Table 1: At many places symbol of element and other places full name have been used, make it uniform. It would be better if content of the table is categorized as per the application viz. Heavy metal, Oil, Hydrocarbons etc. When the manuscript is dealing with heavy metals, it would be better if the content is focussed on heavy metals only.
  • A table can be inserted in the section 3 that can highlight important genes, proteins and metabolites from important studies. It would attract the readership.
  • Please check the manuscript for a few typo, double spacing and grammatical errors
  • A few abbreviations can be used for repetitive words.

Author Response

Reviewer 2

The manuscript, ‘Trends in Harnessing Plant Endophytic Microbiome for heavy 2 metal mitigation in plants: A perspective’ is well written, compiled, and informative. It has included crucial and contemporary aspects of Heavy metal tolerance in the plants conferred by endophytes. The manuscript is worth publishing however a few minor comments are being made, author may consider including:

  • Table 1: At many places symbol of element and other places full name have been used, make it uniform. It would be better if content of the table is categorized as per the application viz. Heavy metal, Oil, Hydrocarbons etc. When the manuscript is dealing with heavy metals, it would be better if the content is focussed on heavy metals only.

Response-We thank the reviewers for their valuable suggestions. The table was revised to focus exclusively on heavy metals only, other sections were removed. Please see the revised manuscript for changes. Also, only the abbreviated forms of the elements were used in the revised manuscript for uniformity.

  • A table can be inserted in the section 3 that can highlight important genes, proteins and metabolites from important studies. It would attract the readership.

Response-The authors agree with the suggestion. A new table 2 discussing the use of different high-throughput technologies and omics biology in the identification of important genes, proteins and metabolites was included in the revised manuscript, please refer for the changes.

  • Please check the manuscript for a few typo, double spacing and grammatical errors

Response-The manuscript was extensively revised for English, typo errors, double spacing and grammatical errors. Also, several sentences were rewritten and revised for English language, please refer to the revised manuscript for changes.

  • A few abbreviations can be used for repetitive words.

Response-We thank the esteemed reviewers for their valuable comments. For the repetitive words, abbreviated forms were used, please see the revised manuscript for changes.

The esteemed Editor and reviewers are acknowledged for their valuble/critical comments made on the manuscript. We surely hope that the revised manuscript has been extensively improved and holds better prospects for consideration in Mdpi Plant journal.

Round 2

Reviewer 1 Report

The authors have made careful revision on the manuscript, I recommend its potential publication in Plants as is.